# 18 year record of circum-Antarctic landfast sea ice distribution allows detailed baseline characterisation, reveals trends and variability

Alexander D. Fraser[1], Robert A. Massom[2,1], Mark S. Handcock[3], Phillip Reid[4,1], Kay I. Ohshima[5], Marilyn N. Raphael[6], Jessica Cartwright[7], Andrew R. Klekociuk[2,1], Zhaohui Wang[8], and Richard Porter-Smith[1]

[1]Australian Antarctic Program Partnership, Institute for Marine and Antarctic Studies, University of Tasmania, Hobart, Tasmania 7001, Australia
[2]Australian Antarctic Division, Channel Highway, Kingston, Tasmania 7050, Australia
[3]Department of Geography, University of California, Los Angeles, CA 90095, USA
[4]Bureau of Meteorology, 111 Macquarie St, Hobart, Tasmania 7000, Australia
[5]Institute of Low Temperature Science, Hokkaido University, Sapporo 060-0819, Japan
[6]Department of Statistics, University of California, Los Angeles, CA 90095, USA
[7]Spire Global, Inc., Glasgow, G3 8JU, UK
[8]Institute for Marine and Antarctic Studies, University of Tasmania, Hobart, Tasmania 7001, Australia

**Correspondence:** Alexander D. Fraser (Alexander.Fraser@utas.edu.au)

**Abstract.** Landfast sea ice (fast ice) is an important though poorly-understood component of the cryosphere on the Antarctic continental shelf, where it plays a key role in atmosphere-ocean-ice sheet interaction and coupled ecological and biogeochemical processes. Here, we present a first in-depth baseline analysis of variability and change in circum-Antarctic fast-ice distribution (including its relationship to bathymetry), based on a new high-resolution satellite-derived time series for the period 2000 to 2018. This reveals a) an overall trend of -882 ± 824 km$^2$/year (-0.19 ±0.18 %/year); and b) eight distinct regions in terms of fast-ice coverage and modes of formation. Of these, four exhibit positive trends over the 18 year period and four negative. Positive trends are seen in East Antarctica and in the Bellingshausen sea, with this region claiming the largest positive trend of +1,198 ± 359 km$^2$/year (+1.10 ± 0.35 %/year). The four negative trends predominantly occur in West Antarctica, with the largest negative trend of -1,206 ± 277 km$^2$/year (-1.78 ± 0.41 %/year) occurring in the Victoria and Oates Lands region in the eastern Ross Sea. All trends are significant. This new baseline analysis represents a significant advance in our knowledge of the current state of both the global cryosphere and the complex Antarctic coastal system that is vulnerable to climate variability and change. It will also inform a wide range of other studies.

## 1 Introduction

Around Antarctica, landfast or fast ice is a stationary and consolidated form of sea ice which is attached to, and held in place by, the coastline or floating ice shelf fronts (World Meteorological Organization, 1970) and icebergs grounded in waters shallower than approximately 400 m (Giles et al., 2008; Fraser et al., 2012). As such, Antarctic fast ice forms only on the continental shelf,

typically in narrow (50 to 250 km wide) bands adjacent to the coast and/or upstream of protrusions into the westward Antarctic Coastal Current that intercept encroaching (drifting) pack ice (Fraser et al., 2012; Nihashi and Ohshima, 2015). Depending on location, Antarctic fast ice can range in persistence from annual through perennial to multi-decadal (e.g., Massom et al., 2010),
with certain regions being highly variable and breaking out and reforming several times per year (e.g., Massom et al., 2009; Fraser et al., 2012).

Antarctic fast ice is not only a sensitive bellwether of climate change and variability (given its intimate linkage and interaction with the high-latitude ocean and atmosphere, Massom et al., 2009; Fraser, 2011; Aoki, 2017), but its distribution also influences the size of adjacent coastal polynyas (Massom et al., 1998, 2001; Nihashi and Ohshima, 2015; Fraser et al., 2019), affecting
regional rates of sea-ice production, water mass modification and the formation of globally-important Antarctic Bottom Water in certain key locations (e.g., Kusahara et al., 2017; Ohshima et al., 2013). Moreover, recent work has shown the importance of fast ice in mechanically bonding and stabilising vulnerable outer margins of floating glacier tongues and ice shelves (Massom et al., 2018; Massom et al., 2015; Massom et al., 2010), and also in controlling the seasonal dynamics and discharge rate of certain outlet glaciers (Greene et al., 2018). Fast ice is also of major ecological importance as a key breeding habitat for
emperor penguins and Weddell seals (Kooyman and Burns, 1999; Massom et al., 2009), plays a role in structuring shallow coastal benthic ecosystems (Clark et al., 2017) and is a region of high primary productivity (concentrated ice algal growth (Meiners et al., 2018)). Coastal fast ice also constitutes a reservoir of nutrients (de Jong et al., 2013) which can substantially enhance primary production in the coastal zone when released into the water column upon fast-ice breakout/melt (particularly for thick, multi-year fast ice, e.g., Shadwick et al., 2013). Finally, fast ice can either facilitate or impede aviation and station
resupply activities, depending on its location, extent and thickness (The Council of Managers of National Antarctic Programs (COMNAP), 2015). It follows that change and/or variability in Antarctic fast-ice distribution and seasonality have wide-ranging ramifications, and characterisation of where and how fast ice is changing is a high priority.

Accurate, consistent, long-term and year-round time-series mapping of Antarctic fast ice at a high spatio-temporal resolution and on a circumpolar scale requires satellite observation, but is technically challenging (Fraser et al., 2009, 2010; Nihashi and
Ohshima, 2015; Fraser et al., 2020; Kim et al., 2018, 2020; Li et al., 2020). Knowledge of its distribution and trends has been identified as a major gap (Vaughan et al., 2013; Meredith et al., 2019). This has severely limited our understanding of the important coastal icescape and key interactive physical, biological and biogeochemical processes therein. Fraser et al. (2012) released an 8.8 y dataset of East Antarctic fast ice extent from 2000 to 2008, but this dataset has not been updated. Using passive microwave satellite data, Nihashi and Ohshima (2015) subsequently produced a dataset of circum-Antarctic fast ice
extent from 2003 to 2011, but at a relatively coarse resolution of 6.25 km/pixel, and this technique does not detect and include young fast ice (Fraser et al., 2019). Li et al. (2020) recently released a high spatial resolution circum-Antarctic dataset of fast ice covering November only in the years 2006-11 and 2016-17 using synthetic aperture radar (SAR) image analysis. However, since November is a month characterised by regionally-variable fast ice retreat (Fraser et al., 2012), it is inadequate for analysis of long term trends in extent.

As such, detailed circumpolar characterisation of fast ice has not been possible due to the lack of a suitable underlying dataset. This gap has recently been filled by the publication of a new time series of fast ice extent from March 2000 to March

2018 (Fraser et al., 2020). This dataset contains 432 contiguous maps of fast ice extent at a 1 km and 15 day resolution, generated by compositing cloud-free visible and thermal infrared imagery from NASA Moderate Resolution Imaging Spectro-radiometer (MODIS) sensors onboard the Terra and Aqua satellites (Fraser et al., 2009, 2010). The process of generating the

cloud-free composite imagery relies upon the MOD35 cloud mask product (Ackerman et al., 2006), which performs brightness temperature and reflectance-based spectral tests to determine the probability of cloud contamination. This product has limita-tions, especially during polar night (Fraser et al., 2010), and the procedure for composite generation may be improved using machine learning-based techniques such as those demonstrated by Paul and Huntemann (2020). Such improvements may be implemented in a future fast ice product.

Here, we use the newly-released fast ice dataset to perform a first detailed characterisation of circum-Antarctic fast ice distribution, change and variability. We first identify eight distinct regions in terms of fast ice co-variability, which form the basis of the new analysis of fast ice trends around Antarctica. These regions differ from the sectors more traditionally used in Antarctic sea ice analyses (Zwally et al., 1983). We then present the overall extent time series and annual climatology, spatial characterisation of mean fast ice persistence, age and timing of minimum/maximum extent across the 18 year dataset. We

also analyse fast ice persistence in concert with bathymetric depth, and interpret this regionally, to more widely assess and determine the linkages between fast ice and grounded icebergs, which act both as stable anchor points for fast ice formation (e.g., Massom et al., 2009; Li et al., 2020) and to intercept and retain encroaching pack ice, thus encouraging fast ice formation upstream (Massom et al., 2001; Massom, 2003).

## 2   Datasets and methods

15 day temporal resolution fast ice maps were obtained from a recently-published NASA Moderate Resolution Imaging Spec-troradiometer (MODIS)-derived 18 year record of Antarctic fast ice extent (Fraser et al., 2020). This dataset consists of 432 con-tiguous maps of fast ice extent at a 1 km spatial resolution. This dataset is freely available at http://dx.doi.org/doi:10.26179/5d267d1ceb60c. In this dataset, fast ice maps were constructed following a semi-automated method whereby persistent edges over a 15 day pe-riod were taken to be the fast ice edge. Manual intervention was required for times and regions where cloud cover persisted

throughout the 15 day window. However, semi-automation was achieved, with 58% of fast ice edge pixels able to be automati-cally retrieved, marking an advance over earlier, more subjective large-scale fast ice maps (e.g., Fraser et al., 2012).

    To underpin definition of regions of fast ice co-variability, fast ice anomaly time series are produced for each $1/4°$ of longitude by subtracting the observed $1/4°$ total fast ice from its repeating climatological cycle. Fast ice regions are defined by performing a spatial cross-correlation of these $1/4°$ longitude fast ice anomaly time series. Nearby regions exhibiting similar

anomaly co-variability are indicated by positive correlation "pockets", and these are grouped manually to define regions. We aim to select regions with high correlation within the region and low correlation outside of the region. The final step of manual regional selection a) allows grouping of fast ice regions across gaps; and b) avoids excessive partitioning of broader regions. Region selection using a decorrelation length scale minimum-based approach to region delineation, as in Raphael and Hobbs

(2014), was unable to be implemented due to extensive coastal regions with no fast ice. This selection of regions of fast ice co-variability is detailed in Appendices A and B.

Fast ice persistence distribution is characterised by calculating the fraction of the time series which each pixel is covered by fast ice, after Fraser et al. (2012). This "per-pixel" mapping is also exploited to visualise per-pixel a) timing of minimum and maximum fast ice extent, b) fast ice age and c) trends in extent. The per-pixel trend map is constructed by fitting a linear trend to each pixel's 18 year time series of extent, and plotting the slope (trend) for each pixel. The map of timing of minimum/maximum fast ice extent is constructed by fitting a Fourier series (first four Fourier components) to the time series of fast ice extent for each pixel. The Fourier parameters are chosen by Levenberg-Marquardt least-squares minimisation (Markwardt, 2009), implemented here due to speed of execution over the large dataset. The resulting timing of minimum/maximum extent is then extracted from the Fourier fit. Here, we prefer to display timing information in this "day of minimum/maximum" format rather than the traditional maps of "day of advance/retreat" used in other sea ice seasonality studies (e.g., Massom et al., 2013) due to the event-based formation and breakout of fast ice, in contrast to the more fine-grained advance/retreat of sea ice. The mean fast ice age map is constructed by calculating the mean time between fast ice formation and subsequent breakout.

We characterise the distribution of fast ice over bathymetry of varying depth by constructing 2D probability distribution functions of International Bathymetric Chart of the Southern Ocean (IBCSO, Arndt et al. (2013))-derived bathymetric depth (50 m bins) vs fast ice persistence (5 % bins). We use this circum-Antarctic bathymetry compilation despite the caveat that all such compilations suffer from a scarcity of data in fast ice-infested waters, owing to a lack of shipboard sonar measurements (Smith et al., 2021). We retrieve the modal value for each persistence bin, then compute the persistence-weighted mean of these modal values to characterise modal formation depth on a circumpolar basis, as well as within the regions defined here.

We also use sea ice concentration from the National Oceanic and Atmospheric Administration/National Snow and Ice Data Center Climate Data Record of Passive Microwave Sea Ice Concentration, Version 3 (Peng et al., 2013; Meier et al., 2017), to compare timing of sectoral fast ice extent to that of overall sea ice. For this timing comparison, we exploit a new technique to model the seasonal cycle of both sea ice and fast ice presented by Handcock and Raphael (2020). This technique, which models each year's annual cycle as a smoothed spline plus a smoothed trend over time, allows daily-resolution calculation of timing statistics even when the input dataset has a bi-weekly resolution, thus facilitating a robust timing comparison between fast ice and overall sea ice extent. The method can estimate the smooth cyclical spline based on an arbitrary and/or irregular data interval. We treated the fast ice extent value as if it was a point measurement on the day at the midpoint of the 15 day cycle. For example, if the start day-of-year was 61 and the end day-of-year was 75, we modeled it as if we had a single measurement at day-of-year $(61 + 75)/2$. Here, we use the invariant cycle described by Handcock and Raphael (2020), rather than the more complex amplitude and/or phase modulated cycles, since we describe the climatological average rather than individual years.

Circumpolar and regional fast ice trends are computed by calculating the regional total fast ice extent, computing the climatological cycle, removing the climatological cycle from the observed totals to form the regional anomalies, and fitting a linear trend to the anomalies in each region. Trend confidence is determined by calculating 95% confidence intervals using the t-distribution. In the calculation of trends, pixels experiencing ice shelf retreat or advance during the 18 year period are removed from this calculation to remove the strong trend contributions caused by these processes.

## 3   Results

### 3.1   Climatological patterns

For the analyses in the following sections, we consider only the eight fast ice regions (as detailed in the Appendices A and B), plus circumpolar total statistics. Fig. 1 shows the total circumpolar fast ice extent time series (a) and its climatological annual cycle (b). A strong annual cycle is evident, with a relatively broad maximum ($\sim$ 601,000 km$^2$) occurring throughout late winter/early spring (day-of-year 273; late September on average), and a well-defined minimum in March ($\sim$221,000 km$^2$, day-of-year 71; mid-March). This indicates that fast ice experiences a seasonal approximate threefold increase in extent. As with overall sea ice (Eayrs et al., 2019; Parkinson, 2019; Simmonds and Li, 2021), fast ice displays an asymmetrical annual cycle, experiencing on average $\sim$ 7 months of advance and 5 months of retreat. As such, fast ice as a percentage of overall sea ice area (extent) varies between a maximum of 12.8 % (8.5 %) in early-mid February (coinciding with the overall sea ice minimum in early-mid February) and around 4.0 % (3.2 %) throughout the winter (mid July to late November). The largest fast ice contribution is from East Antarctica, with the Western Indian Ocean, Eastern Indian Ocean and Australia regions together contributing over half of all fast ice in terms of areal coverage despite only covering 119° of longitude (Table 1).

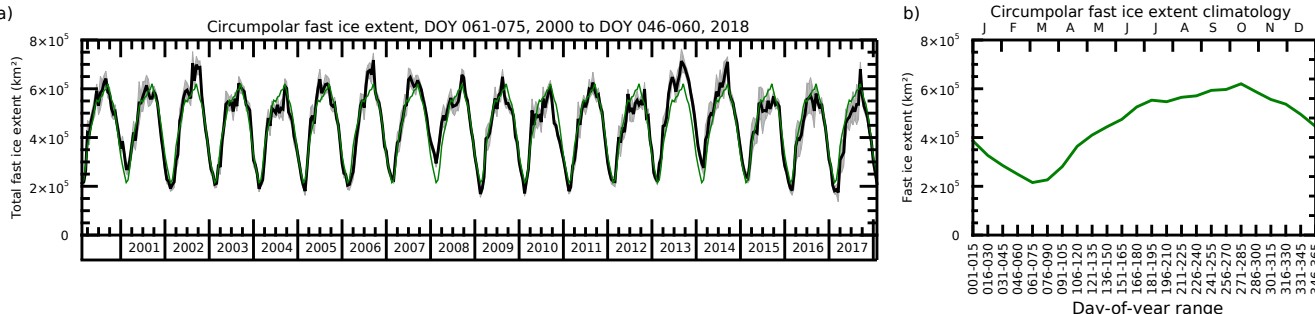

**Figure 1.** a) Bold black line: circumpolar total fast ice extent time series, Mar 2000 - Mar 2018. The gray shading indicates the uncertainty in the underlying dataset, and the thin green line shows the repeating annual cycle. b) Climatological cycle of total fast ice extent by day of year (i.e., the same as the repeating green line in panel a, but with expanded temporal scale).

Fig. 2 shows the circumpolar fast ice persistence distribution as a percentage of time covering each pixel, averaged across the 18 year dataset. This highlights three broad-scale characteristics: a) widespread regions of intermediate persistence (<$\sim$75%, indicating considerable seasonal growth and decay); b) a zonal width of typically $\sim$ 50 - 100 km, but up to 250 km (occurring east of the Mertz Glacier Tongue, $\sim$145° E); and c) localised regions of near-100% persistence likely corresponding to multi-year fast ice (which is mapped later in the analysis of fast ice mean age).

Table 1 indicates that both minimum and maximum fast ice extent (day-of-year 71 and 273, respectively) occur later than the corresponding timings for overall sea ice (comprising both pack and fast ice), which are day-of-year 50 (mid-February) and 264 (mid-September), respectively. Regionally, the result of later fast ice minimum is consistent across all regions, however, a later fast ice maximum only occurs in five of eight fast ice regions (although this may be a consequence of this considerable regional

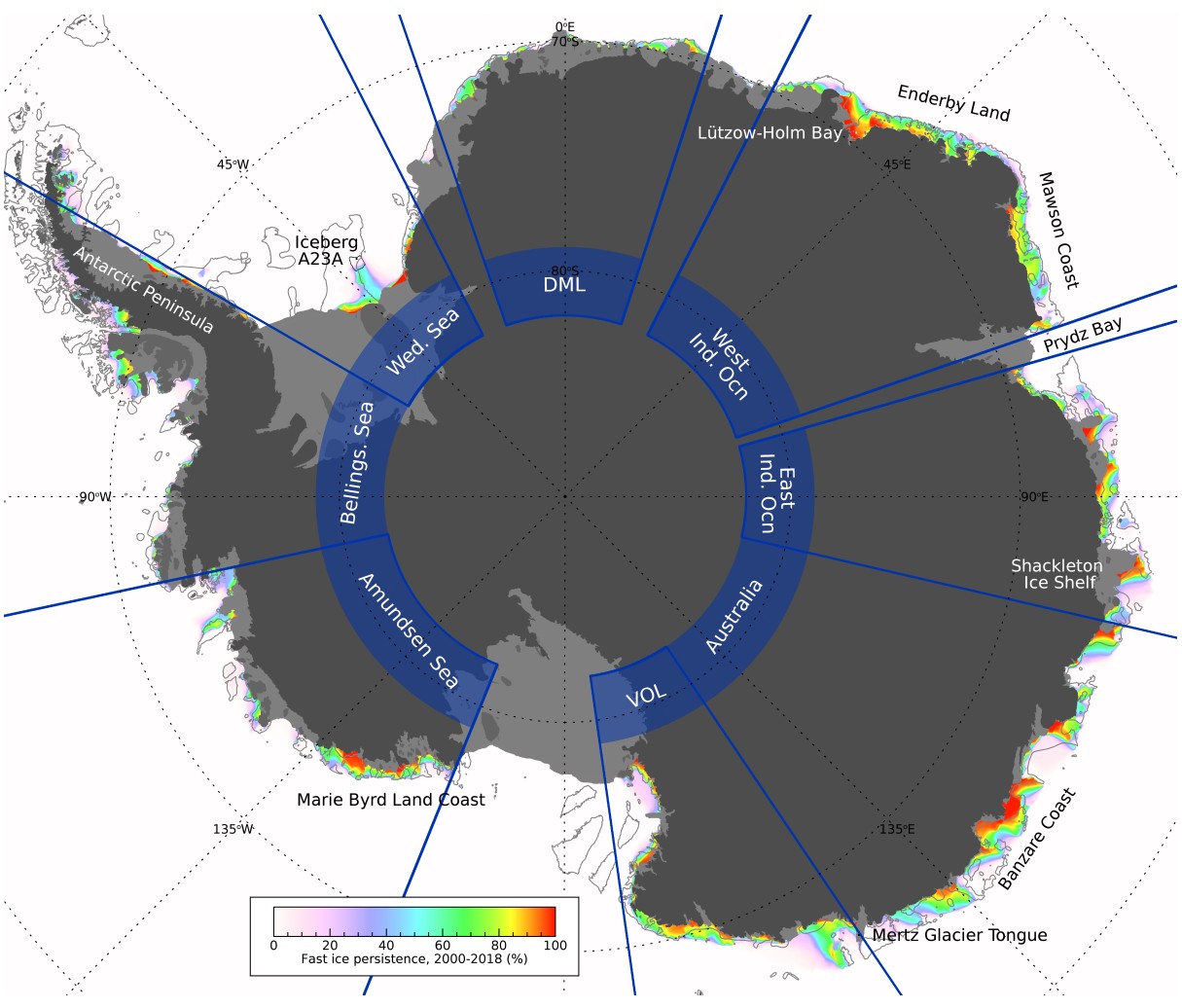

**Figure 2.** Distribution of fast ice persistence, expressed as a percentage of time covering each pixel from Mar 2000 to Mar 2018. Fast ice region boundaries are indicated by blue radial lines. Gaps between regions occur in areas of minimal fast ice coverage. The 403 m isobath, corresponding to the weighted-mean circumpolar fast ice depth, is indicated as a gray contour. DML is Dronning Maud Land and VOL is Victoria and Oates Lands. The early-2000 position of large tabular, grounded iceberg A23A is indicated here in black outline, in the southern Weddell Sea.

| Region | Fast ice: Extent ($10^3$ km$^2$) and DOY (brackets) at | | Sea ice: Extent ($10^3$ km$^2$) and DOY (brackets) at | | Fast ice trend and confidence interval (%/year) |
|---|---|---|---|---|---|
| | Minimum | Maximum | Minimum | Maximum | |
| Circumpolar | 221 (71) | 601 (273) | 3,213 (50) | 18,900 (264) | -0.19 ± 0.18 |
| Dronning M. L. (19° W - 18° E) | 4.7(73) | 21 (294) | 99 (51) | 3,122 (269) | 1.80 ± 0.47 |
| West Ind. Ocn (27° E - 71° E) | 33 (72) | 111 (264) | 134 (58) | 2,252 (290) | 0.41 ± 0.30 |
| East Ind. Ocn (74° E - 103° E) | 24 (74) | 72 (275) | 122 (54) | 1,362 (265) | -1.38 ± 0.43 |
| Australia (103° E - 146° E) | 53 (70) | 139 (260) | 173 (53) | 1,112 (275) | 1.10 ± 0.35 |
| Vict. Oates L. (146° E - 172° E) | 31 (67) | 85 (285) | 312 (49) | 1,118 (230) | -1.78 ± 0.41 |
| Amundsen Sea (102° W - 158° W) | 36 (66) | 66 (246) | 581 (51) | 2,766 (258) | -2.00 ± 0.45 |
| Bellings. Sea (60° W - 102° W) | 23 (79) | 64 (275) | 237 (65) | 1,244 (242) | 2.81 ± 0.50 |
| Weddell Sea (27° W - 60° W) | 20 (56) | 41 (280) | 1,164 (50) | 2,768 (241) | -2.59 ± 0.69 |

**Table 1.** Mean extent (in $10^3$ km$^2$) and mean timing (day-of-year, or DOY) of minimum/maximum extent of both fast ice (left) and overall sea ice (centre) for the entire continent and the eight fast ice regions used here. The right column gives the trend in fast ice extent across the 18 year dataset. Dronning M. L. is Dronning Maud Land, West Ind. Ocn is Western Indian Ocean, East Ind. Ocn is Eastern Indian Ocean, Vict. Oates L. is Victoria and Oates Lands, and Bellings. Sea is Bellingshausen Sea.

variability in the timing of overall sea ice extent maximum). Maps of timing of minimum and maximum fast ice extent are presented to provide a more localised context for regional studies involving fast ice (Fig. 3). These reveal remarkable differences within neighbouring areas. For example a) Enderby Land fast ice (39° to 52° E) achieves a minimum in April whereas along the Mawson Coast (55° to 71° E) this occurs in February-March; and b) fast ice on the western side of the Antarctic Peninsula reaches minimum later (April) than that on the eastern side (February-March). Regional changes in maximum extent timing are more variable than minimum extent timing, likely due to the broad fast ice peak in the climatological cycle (Fig. 1b). No latitudinal gradient is apparent in either timing metric.

The map of fast ice mean age is presented in Figure 4a. Regions of multi-year fast ice (i.e., mean age >12 months) are typically located either east (upstream) of physical barriers to the westward drift of pack ice in the Antarctic Coastal Current, within deep, sheltered embayments (e.g., Lützow-Holm Bay, ∼40° E), or adjacent to coastal flaw leads indicating the presence of a shear zone (especially in the southern Weddell Sea). As such, the majority is in East Antarctica (e.g., along the Banzare Coast, ∼130° E), although major areas are found throughout the Weddell Sea region and along the Marie Byrd Land Coast (∼134 - 159° W). Multi-year ice is generally synonymous with highly-persistent fast ice (i.e., red pixels corresponding to near-100% persistence in Fig. 2), however we find limited regions where relatively low-persistence fast ice (e.g., persisting through 80% of the year on average) can be multi-year (e.g., north of the Wilkins Ice Shelf, ∼74° W), indicating a change to the annual cycle throughout the 18 year study period.

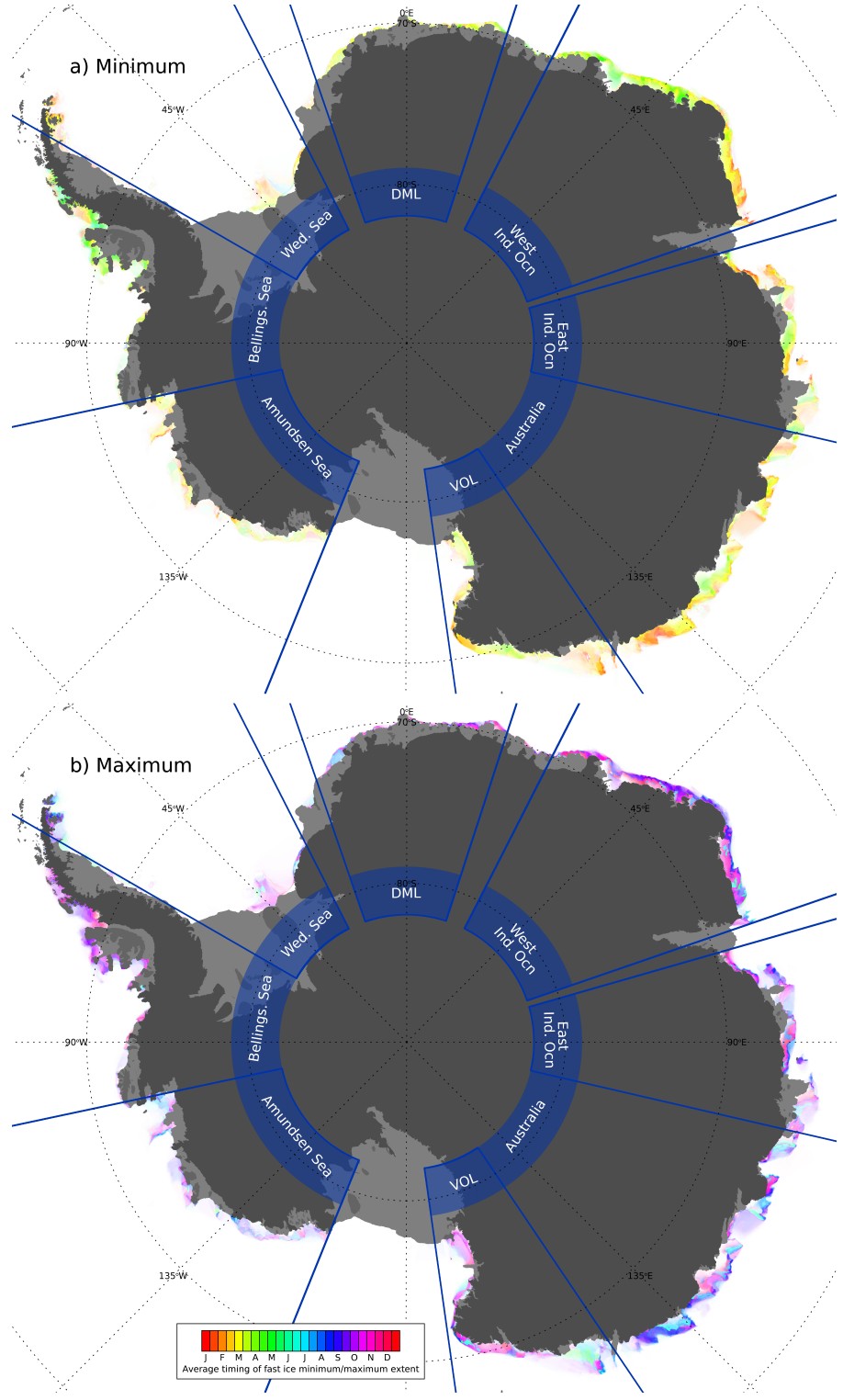

**Figure 3.** Per-pixel average timing of minimum (a) and maximum (b) fast ice extent. For each pixel, color saturation is weighted by annual cycle magnitude (i.e., pixels with persistent multi-year fast ice or no fast ice are transparent.)

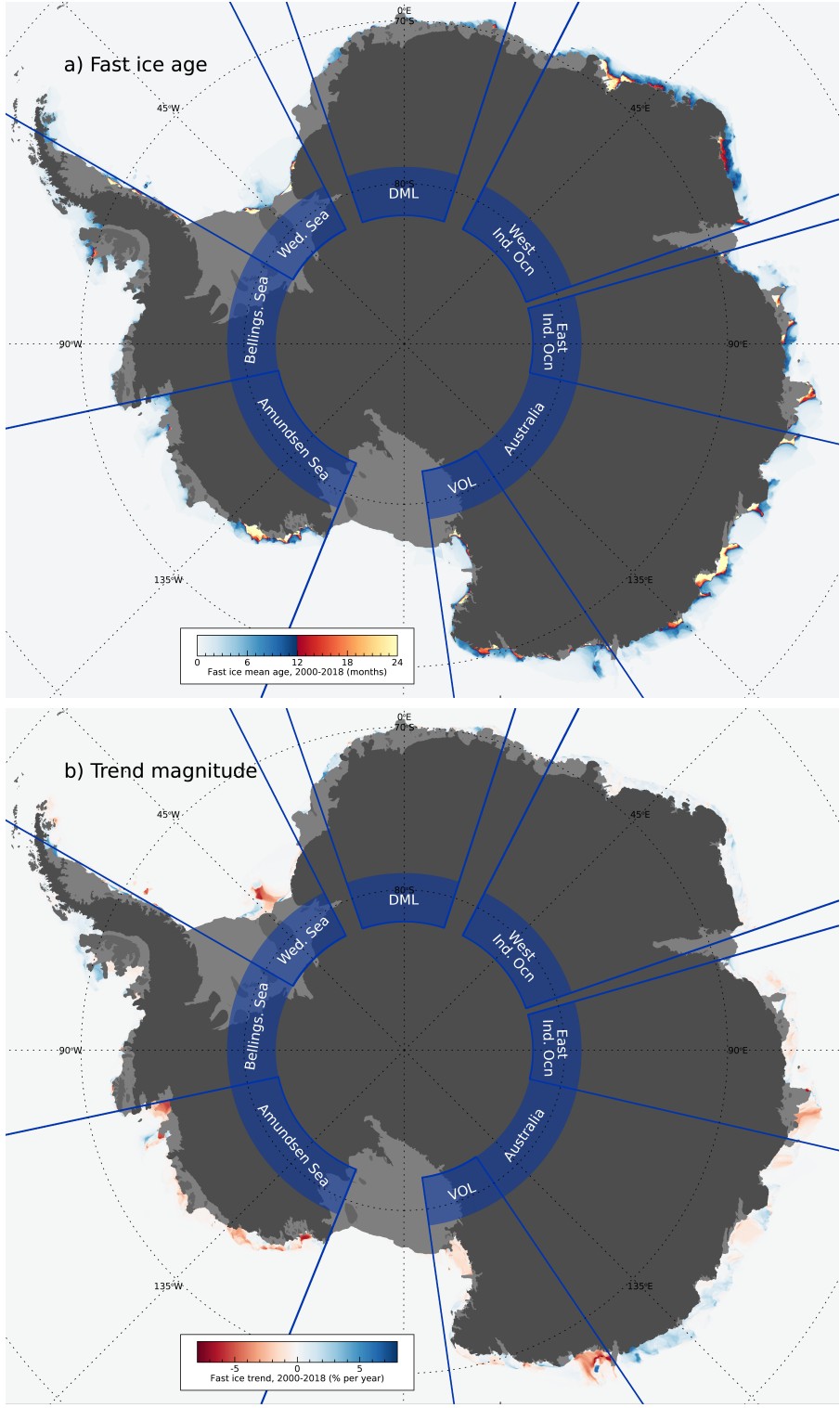

**Figure 4.** a) Mean fast ice age over the 18 year study period. White to blue pixels indicate seasonal fast ice. The color break from a blue hue to a red hue at 12 months is chosen to highlight regions of multi-year fast ice (red-yellow hues). b) Per-pixel trends in fast ice extent (%/year).

## 3.2 Fast ice extent anomalies and linear trends

The circumpolar total anomaly time series (Fig. 5a) indicates the presence of extended periods with only transient departures from the climatological mean (e.g., 2000 to mid-2007; mid-2008 to 2012), as well as three more persistent departures: extended positive anomalies occurring in mid-2007 to mid-2008; throughout 2013 and 2014; and a negative anomaly from mid-2015 onwards. A marginally-significant (p≃0.04) negative trend of $-882\pm823$ km$^2$/year (-0.19 $\pm$ 0.18 %/year) is reported across Antarctica for the 18 year time series. As shown in Fig. 5b-i and Table 1, all regions exhibit statistically-significant trends, further indicating that the methodology for defining partitions is robust, even at an inter-decadal time-scale. Four regions show positive trends (Dronning Maud Land, Western Indian Ocean, Australia and Bellingshausen Sea regions), with the remainder negative. Strong and opposing trends are frequently observed to occur in adjacent regions indicating that this partitioning supports preservation of these opposing regional signals.

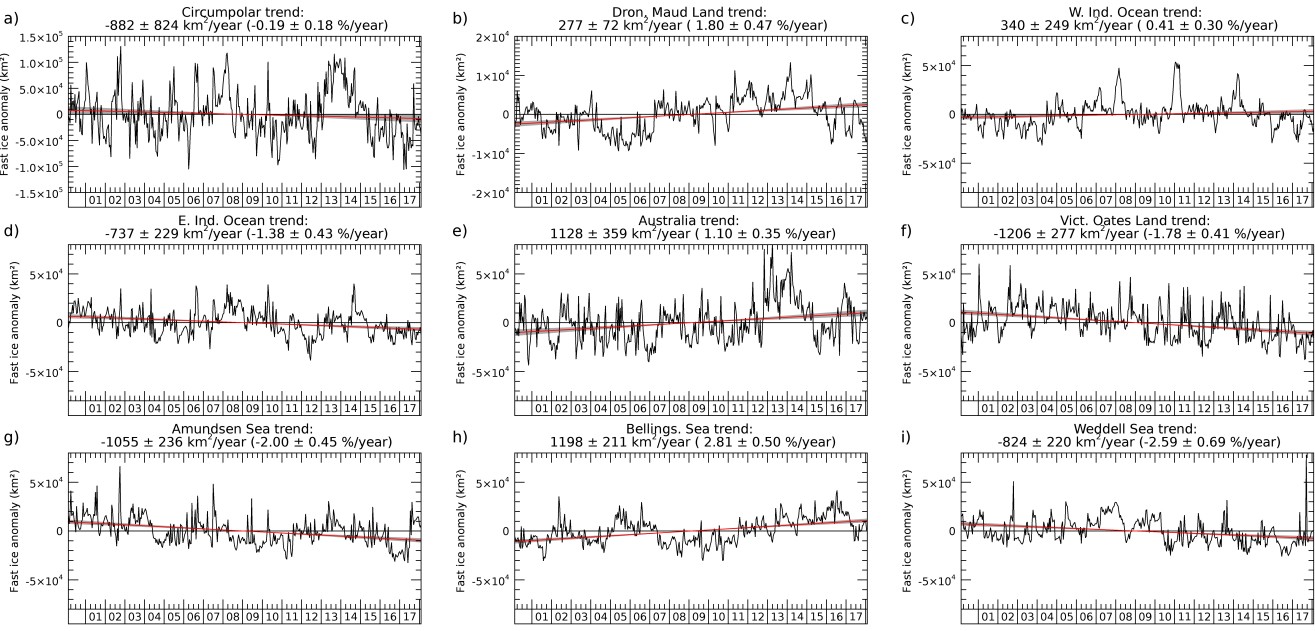

**Figure 5.** Fast ice extent anomalies (black line) and linear trends (red line) for the eight fast ice regions. Trend magnitude and bootstrapped 95% confidence interval are indicated in the title of each sub-plot. Note the y-axis scale is the same for all plots except panels a and b. All trends are significant, with a p-value of ≃0.00 for all trends except for the circum-Antarctic (panel a; p≃0.04) and Western Indian Ocean (panel c; p≃0.01) regions.

Per-pixel trends in fast ice extent (Fig. 4b) in excess of $\pm8$ %/year are observed, corresponding to pixels exhibiting an extreme change in fast ice cover across the study period. These occur in areas exhibiting major icescape change, e.g., both upstream and downstream of the Mertz Glacier Tongue ($\sim144°$ E), which calved in 2010 (Fogwill et al., 2016), and in the Weddell Sea between the Ronne-Filchner Ice Shelf and grounded iceberg A23A, which has gradually drifted northward (Li et al., 2020). Broader regions of similar signed (but weaker) trend are also apparent, e.g., the positive trend across much of the

western half of East Antarctica (i.e., the Dronning Maud Land and Western Indian Ocean regions), indicating a consistent fast ice response to environmental forcing over a large spatial scale. Given the widespread distribution of the positive trend along the eastern part of the Weddell Gyre, we speculate that this environmental association is likely oceanic in nature. We also note that this is a region of increasing summertime, springtime and wintertime sea ice concentration (Fig. 2 of Simmonds and Li, 2021), which may favour formation of more extensive or longer duration of fast ice coverage, i.e., this fast ice trend may be associated with an oceanic trend which has atmospheric drivers.

## 3.3 Bathymetric controls on fast ice distribution

Analysis of the bathymetric distribution of fast ice provides fundamental knowledge on the formation mode of fast ice (i.e., iceberg-associated vs formation within embayments). Circumpolar fast ice persistence as a two-dimensional histogram, binned by bathymetric depth is given in Fig. 6. This analysis indicates that the weighted (by persistence) mean depth of fast ice persistence occurs at ~403 m, in line with earlier estimates linking fast ice extent with grounded icebergs in East Antarctica (Massom et al., 2001). The 403 m bathymetric contour is shown on Figure 2 as a solid grey line. This isobath only bears visual resemblance to areas containing persistent fast ice for much of East Antarctica ($20°$ W to $172°$ E) and the Ross Sea ($172°$ E to $130°$ W), indicating regional variability in either the actual iceberg grounding depth or the reliance of fast ice on the stability provided by grounded icebergs. Fig. C1 shows persistence-weighted histograms of fast ice formation depth for the eight fast ice regions. Remarkable variability is observed across the regions, with the weighted mean of modal bathymetric depth ranging from >420 m (four contiguous regions from the Eastern Indian Ocean to the Amundsen Sea) to as shallow as ~ 200 m (Bellingshausen Sea region), confirming that the ~400 m isobath is only useful as an indicator of fast ice propensity for certain regions.

## 4 Discussion

### 4.1 Fast ice distribution, age and trends

Overall, a significant and negative trend was found in circumpolar total fast ice extent (-882 $\pm$ 823 km$^2$/year or -0.19 $\pm$0.18 %/year, Fig. 5a). When partitioned into appropriate regions, opposing trends are observed in most neighbouring regions (Fig. 5b-i): positive trends in the Dronning Maud Land, Western Indian Ocean, Australia and Bellingshausen Sea regions; negative elsewhere. Fraser et al. (2012) found that a significant positive trend was observed in the Indian Ocean sector (20 - $90°$ E) from March 2000 - Dec 2008. Here, we find that this trend, largely represented by the Western Indian Ocean region, did not persist after 2008 (see Fig. 5c). Furthermore, persistent (e.g., 12 months or greater duration) "events" are evident in many anomaly plots (e.g., Fig. 5e, exhibiting a positive anomaly persisting from 2013 to 2014 in the Australia region, and contributing to the positive circumpolar total anomaly at the same time, Fig. 5a). Investigation of drivers of both regional trends and significant events within these regions is planned for the future. We note that the time series of circumpolar fast ice anomaly (Fig. 5a) bears close resemblance to that of overall sea ice extent for the same time period (time series given in Fig. 2B of Parkinson,

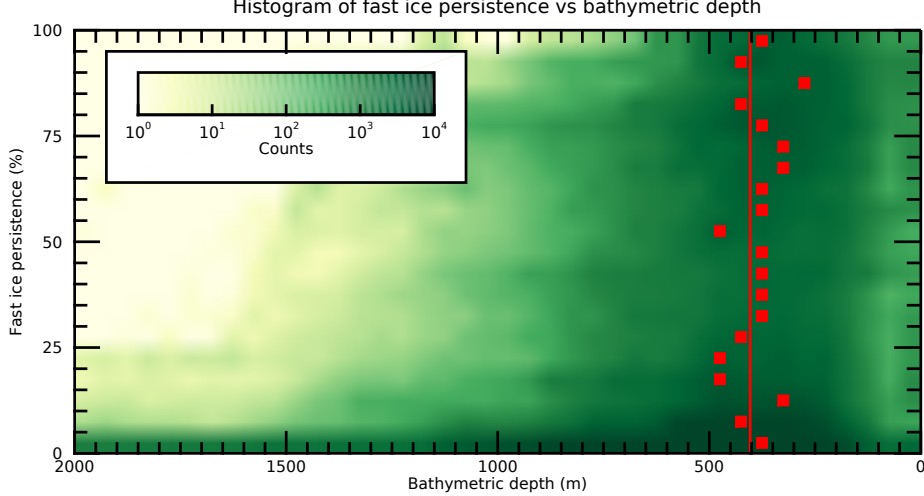

**Figure 6.** Two-dimensional histogram of circumpolar fast ice persistence (y-axis) versus bathymetryic depth (x-axis, taken from the International Bathymetryic Chart of the Southern Ocean (Arndt et al., 2013)). For each fast ice persistence bin (5% wide bins), the most frequent (modal) depth value is highlighted with a red square. The persistence-weighted mean of these modal values is indicated with the red vertical line, here corresponding to a bathymetryic depth of ∼403 m.

2019), with positive anomalies in 2007/08 and 2013/14, with a decline from 2014-2017. This association will also be explored more comprehensively in future work. We also note that regions experiencing positive fast ice trends also coincide with regions experiencing trends toward higher sea ice concentration. While the time periods of the Simmonds and Li (2021) paper differ to that presented here, we can indeed see that the general distribution of positive trend coincides with the southern hemisphere trend in concentration from 1979 to 2020 in Sept-Nov (Fig. 2B, bottom row of Simmonds and Li, 2021), but this link is somewhat less convincing for other seasons. Nearby pack ice concentration is thought to buffer fast ice against wave-induced breakout (Crocker and Wadhams, 1989), and may also indicate that common environmental forcing is favourable for sea ice formation/preservation. That this relationship is strongest is September to November is not surprising, since this season coincides with maximum fast ice extent.

We find here an approximately threefold difference between maximum and minimum fast ice extent, with a minimum occurring in mid-March and a maximum in late September. The circumpolar seasonal cycle is much lower in amplitude than that of overall sea ice extent (with a wintertime maximum extent nearly six times higher than its summertime minimum, (Eayrs et al., 2019; Parkinson, 2019)), a likely manifestation of the relatively large portion of fast ice which is multi-year combined with the limit of maximum fast ice extent imposed by the distribution of grounded icebergs.

## 4.2 Timing of maximum and minimum extent

The circumpolar fast ice cycle is delayed relative to that of overall sea ice, with the fast ice minimum (maximum) occurring 21 (nine) days later than that of sea ice (shown in Table 1), in agreement with the findings of Fraser et al. (2012). In the absence of particular case studies, which are out of scope here, we speculate that this lag may be due to one or more of the following reasons: a) adjacent pack ice may act as a protective buffer against dynamically-induced breakout (e.g., swell may be attenuated by adjacent pack, protecting the fast ice (Ushio, 2006)); b) the presence of pack ice at the fast ice edge may reduce "mode-3" summertime solar heating of the surface water (Jacobs et al., 1992), leading to lower basal melt rates under the fast ice (Arndt et al., 2020) and higher mechanical strength (Fedotov et al. (1998) estimate only 20-30% of wintertime flexural fast ice strength remains by the time basal melt becomes widespread); or c) fast ice is simply able to persist longer into the summer due to the inherent shelter afforded by its formation within certain embayments (e.g., Lützow-Holm Bay, $\sim$40° E, (Ushio, 2006; Aoki, 2017)).

Regarding timing of maximum extent, Fraser et al. (2012) found that fast ice maximum occurs earlier than the overall sea ice maximum in the Indian Ocean and the Western Pacific Ocean sectors (covering much of East Antarctica). We find here that this result holds for only three of eight regions (Table 1), and that the timing in maximum extent is far more regionally variable (range: 48 d) than that of minimum extent (range: 13 d), a result also indicated spatially in Fig. 3. This is likely related to the bathymetric limit imposed on maximum fast ice extent, i.e., fast ice coverage around the outermost grounded icebergs is generally achieved by midwinter, and limited further growth occurs only upstream of obstacles to the coastal current, until September. Such growth is likely stochastic and event-based, imparting variability to the timing of maximum extent.

Based on a dataset covering the years 2006–2011 and 2016–2017, Li et al. (2020) found a mean November extent of $\sim$495,000 km$^2$, which is much lower than the mean maximum extent found here ($\sim$601,000 km$^2$). However, we have shown that November is after maximum fast ice extent in every region, so we suggest that circumpolar studies of maximum fast ice extent are best conducted around late September. Full consideration of the seasonality (i.e., timing of formation, breakout, and presence duration; and change in these quantities) of fast ice is outside of the scope of this paper, however complex regional patterns have been identified in an analysis of overall East Antarctic sea ice seasonality (Massom et al., 2013), so future work on this is planned using the fast ice dataset (Fraser et al., 2020).

## 4.3 Bathymetric controls on fast ice distribution

We have shown large regional variability in the formation depth of fast ice, ranging from $\sim$200 m to $\sim$450 m (Fig. C1). Such regional variability has not been identified in earlier work (e.g., Massom et al., 2001; Fraser et al., 2012). This regional dependence on bathymetry may also suggest fundamental regional differences in fast ice formation mode. As an example of this, consider the distribution of fast ice persistence by depth in the Bellingshausen Sea (Fig. C1, cyan line). In this region, relatively few grounded icebergs exist (Figure 4 in Li et al. (2020)), so fast ice predominantly forms between coastal margins (including islands), and is known as "regime 1" fast ice (Fraser et al., 2012). By contrast, in the Eastern Indian Ocean and Australia regions, "regime 2" fast ice predominates (Fraser et al., 2012), and a close relationship is found between grounded

icebergs and persistent fast ice (Li et al., 2020). In East Antarctica, icebergs have been observed to ground at depths up to around 400 m (Massom, 2003; Massom et al., 2009), although there is evidence for deeper grounding (in excess of 500 m) in some regions of the East Antarctic continental shelf (Beaman and Harris, 2005), and indeed newly-calved icebergs with keels of up to 600 m are known to calve from fast flowing outlet glaciers (Dowdeswell and Bamber, 2007). More detailed understanding of the mechanism of fast ice formation, as provided here, is crucial for development of the next generation of

prognostic regional fast ice models which require tuning of tensile strength (Lemieux et al., 2016).

## 4.4 Future work

Massom and Stammerjohn (2010) discussed a future scenario in which an increase in iceberg discharge from Antarctic ice shelves results in an increase in fast ice extent. In light of our analysis of fast ice persistence in the context of bathymetry, we suggest that this increase would occur only in those regions where few icebergs are currently available to ground on the shallow

bathymetry, i.e., the Bellingshausen Sea and Dronning Maud Land regions, as well as continental shelf shoals in the central Ross and Weddell seas (see 403 m bathymetric contour on Figure 2). Most other regions may already have sufficient density of grounded icebergs to act as fast ice anchors, as detailed in (Li et al., 2020). We also consider a future scenario in which the recently-detailed marine ice cliff instability mechanism is initiated (Pollard et al., 2015), whereby glacier/grounding line retreat results in high ($>\sim90$ m) ice cliffs at the glacier terminus, resulting in calving of icebergs with extremely deep (in excess of

800 m) keels. Evidence for this process exists in the form of very deep sediment scours around Pine Island Glacier ($\sim101°$ W, (Wise et al., 2017)), estimated to have occurred in the early Holocene. Presence of such deeply-keeled icebergs around the Antarctic continental shelf would allow the grounding of icebergs in new regions, completely altering the distribution of fast ice. For this reason, the next generation of coupled Antarctic ice/ocean models with fast ice should consider prognostic iceberg calving and grounding. Near-coastal bathymetric data paucity, leading to high uncertainty in current Antarctic bathymetric

compilations, is also a limiting factor for this kind of study, so should be addressed as a priority (Smith et al., 2021). In addition to fields of smaller grounded bergs, it is also worth re-iterating the profound and unpredictable effects that large tabular icebergs can have on regional fast ice extent (e.g., Fogwill et al. (2016)), particularly when grounded for several decades.

Compared to the earlier dataset covering only East Antarctica from 2000 to 2008 (Fraser et al., 2012), this new 18 year time series (Fraser et al., 2020) is a much more comprehensive dataset from which to gauge long-term change in fast ice

extent. However, we note that the more recent dataset is still shorter than the 30 year "climate" threshold defined by the World Meteorological Organisation (Arguez and Vose, 2011). Indeed, the residence times of large grounded icebergs which have profound effects on fast ice distribution can be as long as several decades (e.g., B09B which was grounded at $\sim148°$ E from 1992 to 2010 (Leane and Maddison, 2018); A23A which has been grounded in the central Weddell Sea since 1991, i.e., currently 29 years (Paul et al., 2015)). As such, the 30 year threshold may also be appropriate to apply to Antarctic fast ice, in

order to preclude undue influence of stochastic large iceberg grounding, giving a strong impetus to extend this dataset.

Although our fast ice analysis is circum-Antarctic in extent, performed at a high spatio-temporal resolution and covers 18 years, it is still only limited to extent/distribution. The underlying dataset (Fraser et al., 2020) does not consider other physical fast ice properties, including freeboard/thickness, thickness of overlying snow, roughness or albedo. Complete physical

characterisation of fast ice requires such data. Giles et al. (2008), working with synthetic aperture radar (SAR) imagery of East Antarctic fast ice, indicated that thickness and roughness are likely closely related, ascribing values of 1.7 and 5.0 m thickness to "smooth" and "rough" fast ice, respectively, however this is an overly-simplistic methodology for estimating thickness. Work is underway on addressing this knowledge gap by remotely sensing circum-Antarctic fast ice roughness and thickness from altimetric satellite data.

## 5   Conclusions

Here, using a newly-released, long-term (18 year), high-quality and high-resolution dataset of circum-Antarctic fast ice, we have for the first time:

- Presented the baseline characterisation of fast ice mean persistence, annual cycle, mean age, and timing of minimum and maximum extent;

- Defined eight new fast ice regions based on fast ice anomaly co-variability;

- Determined and discussed fast ice extent trends in these eight regions, revealing marked regional variability in trend (as well as inter-annual variability in fast ice extent within each region); and

- Discussed fast ice characteristics in terms of its links with bathymetric depth, indicating formation modes within each region.

Although this work greatly advances the state of knowledge on Antarctic fast ice distribution and variability, deeper understanding of Antarctic fast ice is still limited by a paucity of studies on the environmental factors driving changes in fast ice extent. One-dimensional thermodynamic studies have indicated the sensitivity of fast ice to environmental drivers, including both the atmosphere and the ocean (e.g., Heil, 2006; Lei et al., 2010; Hoppmann et al., 2015; Brett et al., 2020), however drivers of change in horizontal fast ice distribution are relatively poorly understood. Of the limited studies of fast ice extent formation/breakout, a wide range of potential drivers have been identified (including remote atmospheric teleconnections (Aoki, 2017; Sato et al., 2021), a range of local atmospheric parameters (Fraser, 2011; Zhai et al., 2019; Leonard et al., 2021), swell-induced breakup and anomalous snow cover (Ushio, 2006) and basal melt (Arndt et al., 2020)), however no unifying picture has emerged. Work is planned to use the new circumpolar fast ice dataset (Fraser et al., 2020) in conjunction with datasets of atmospheric and oceanic parameters to address this shortcoming, in order to elucidate such drivers. Due to regionally-specific drivers, we suggest that coupled ocean/sea ice models capable of realistically forming Antarctic fast ice are an important tool for studing fast ice variability, and urgently need to be developed.

*Data availability.* The fast ice dataset analysed here is freely available at http://dx.doi.org/doi:10.26179/5d267d1ceb60c. Sea ice concentration data were obtained from the NOAA/NSIDC Climate Data Record of Passive Microwave Sea Ice Concentration, Version 3, and are available at https://doi.org/10.7265/N59P2ZTG.

## Appendix A: Sectoral anomalies and trends

In the main text, we focus on fast ice characteristics in eight newly-defined regions, as defined in the following section. Here, however, we report the fast ice extent anomaly (observation minus repeated climatological cycle) and linear trend for the five commonly-used oceanic sectors (Zwally et al., 1983), in order to assess their suitability for partitioning fast ice. These are shown in Fig. A1. Two sectors show significant trends: the Ross Sea (-1.43 ± 0.26 %/year) and the Bellingshausen and Amundsen seas (0.67 ± 0.55 %/year) sectors. The remainder are insignificant, which may either be genuine features, or may

indicate inappropriate (for fast ice) region selection (i.e., regions defined in this way may split areas of fast ice which co-vary).

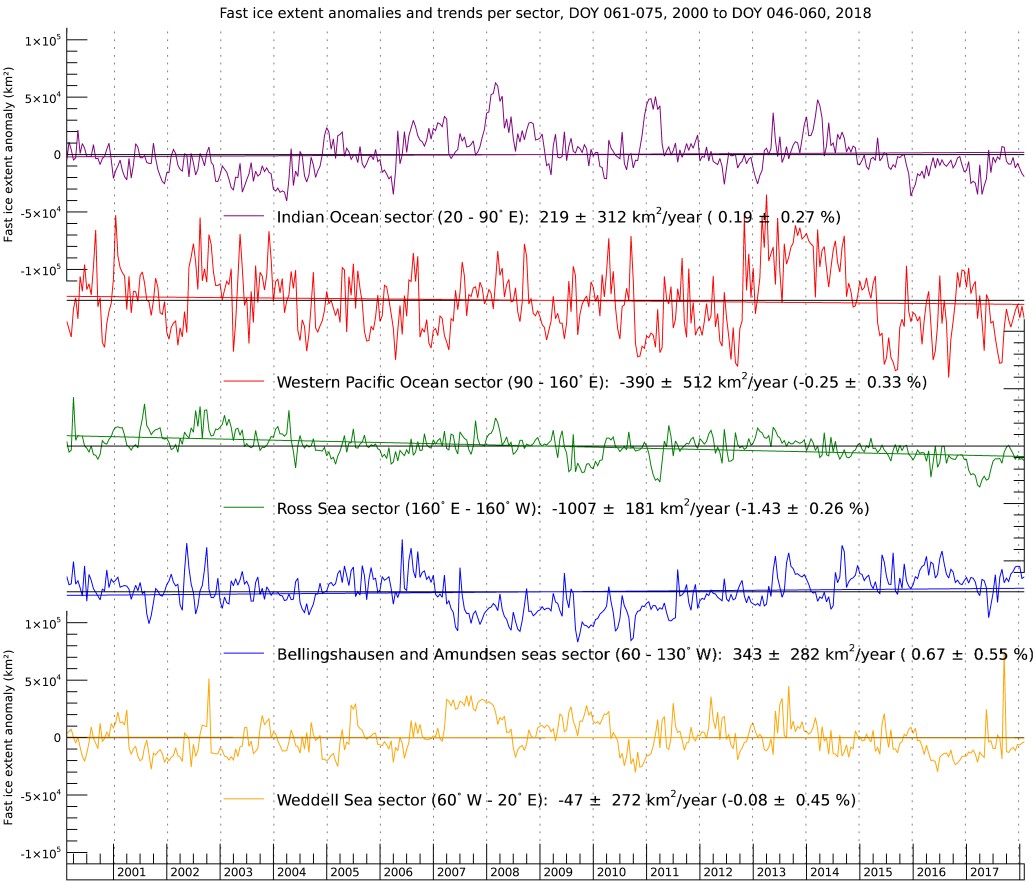

**Figure A1.** Fast ice extent anomaly time series for the 18 year period. Only the Bellingshausen and Amundsen seas and Ross Sea sectors show significant trends.

## Appendix B:  Selection of fast ice regions

To investigate a more appropriate regional split, we perform cross-correlation on 1/4° longitude fast ice extent anomaly time-series (Fig. B1). Regions which co-vary exhibit high cross-correlation. Investigation of the fast ice anomaly cross-correlation matrix as a function of longitude indicates that eight regions are needed to appropriately partition fast ice, indicated as blue boxes. As with the new definition of sea ice sectors in Raphael and Hobbs (2014), this was largely a manual process, constrained by the content in Fig. B1. In the case of Raphael and Hobbs (2014), their selection was guided by functions of sea ice extent standard deviation and decorrelation length scale, i.e., two quantities whose boundaries did not always match spatially, necessitating a subjective decision. Furthermore, Raphael and Hobbs (2014) select boundaries based on local minima of these quantities, however the choice of which local minimum should be selected, when more than one option exists (e.g., the boundary between the Ross and Bellingshausen/Amundsen regions in their Fig 1), is somewhat subjective. This parallels our selection and the subjective elements within. As with Raphael and Hobbs (2014), most section definitions here were quite objective (e.g., the Australia region: Fig B1 shows that this box contains only blue pixels, indicating positive cross-correlation within this region, and is surrounded by red pixels, indicating negative cross-correlation). However, we concede that for more gradually-decorrelating regions (e.g., the demarcation between the Eastern Indian Ocean and Western Indian Ocean regions), the subjective element is higher. We also add a caveat that the region selection process is performed in an *a posteriori* fashion (as in Raphael and Hobbs (2014)), i.e., obtained by analysing regions which co-vary, rather than from physical principals, which may enhance statistical significance of the trends reported here. We suggest that this is appropriate in the absence of detailed large-scale knowledge of physical fast ice formation mechanisms.

Although this fundamental region definition is based on a simple and robust methodology, its implementation and the resulting regional definition in Antarctica require discussion. Firstly, since it is based on the cross-correlation of longitudinal slices, it is unable to separate distinct fast ice areas which share a longitude but differ in latitude. Such cases are encountered around a) 163 - 171° E (Victoria and Oates Land coasts); and b) 60 - 61° W (the eastern side of the Antarctic Peninsula). Although our technique is too simple to account for this longitudinal degeneracy, we note that in both cases, similar trends (Fig. 3b in the main text) are encountered at both areas of fast ice within the longitude zone (i.e., weak negative in the former, and positive in the latter), giving confidence that the region definition is unaffected by this caveat.

Our region selection methodology indicates that the fast ice on the eastern side of the Antarctic Peninsula should be considered a part of the Bellingshausen Sea region, rather than the Weddell Sea region. This is somewhat surprising given the oceanic connection from this region to the rest of the Weddell Sea. However, this ice is much more proximal to the western side of the Antarctic Peninsula than it is to the fast ice in the eastern flank of the Weddell Sea region, indicating that localised atmospheric conditions may be a dominant driver here. This hypothesis is supported by the positive fast ice trend encountered in the Bellingshausen Sea region – a region which has experienced a trend toward cooler surface air temperatures since the late 1990s (Turner et al., 2016; Sato and Simmonds, 2021).

We also consider that the boundary between the Australia and Victoria and Oates Lands regions (at 146° E) may be an artefact of the "ice-scape" regime shift which occurred in the region after the ungrounding of iceberg B09B and subsequent

calving of the Mertz Glacier Tongue in 2010 (Leane and Maddison, 2018). To determine the influence of this event, the regional selection algorithm was re-run using only pre-calving and only post-calving fast ice anomaly data, with the result that the boundary location is correctly located in the pre-calving regime, but the fast ice variability off the Adélie/George V Land coast becomes somewhat more homogeneous in the post-calving regime, as expected following the removal of a major dynamical barrier, with the apparent regime boundary shifting to $\sim$160° E (not shown).

We note here that the region selection is non-conservative (i.e., some longitudes with minimal fast ice coverage are not assigned a region). Without extensive fast ice coverage, these longitudes were unable to be assigned a region. If such regions retain extensive fast ice cover in the future (e.g., in response to major icescape change such as the grounding of a large tabular iceberg) then such longitudes may need to be assigned a region.

## Appendix C: Regional bathymetric constraints on fast ice formation

In addition to the two dimensional histogram of fast ice persistence by bathymetric depth presented in the main text, we present in Fig C1 the projection of this two dimensional histogram onto the x-axis, linearly weighted by persistence (so that high-persistence fast ice contributes more). We also present this histogram for each fast ice region (coloured lines).

*Author contributions.* All authors edited the manuscript. ADF led the analysis, produced all figures, drafted the paper and coordinated co-authors. RAM and KIO provided scientific and editorial feedback, and direction to the project. MSH and MNR provided statistical expertise in the time series analysis. ARK, ZW, PR, JC and RP-S assisted with data analysis, interpretation and visualisation.

*Competing interests.* The authors declare no competing interests.

*Acknowledgements.* This project received grant funding from the Australian Government as part of the Antarctic Science Collaboration Initiative program, and contributes to Project 6 of the Australian Antarctic Program Partnership (Project ID ASCI000002). This work was supported by the Australian Research Council's Special Research Initiative for Antarctic Gateway Partnership (Project ID SR140300001), and contributes to Australian Antarctic Science Project 4116. The authors are grateful to Greg Leonard and one anonymous reviewer for their constructive comments, and to Chad Greene for discussions involving data presentation. Computation and storage facilities were provided by the National eResearch Collaboration Tools and Resources (NeCTAR) project.

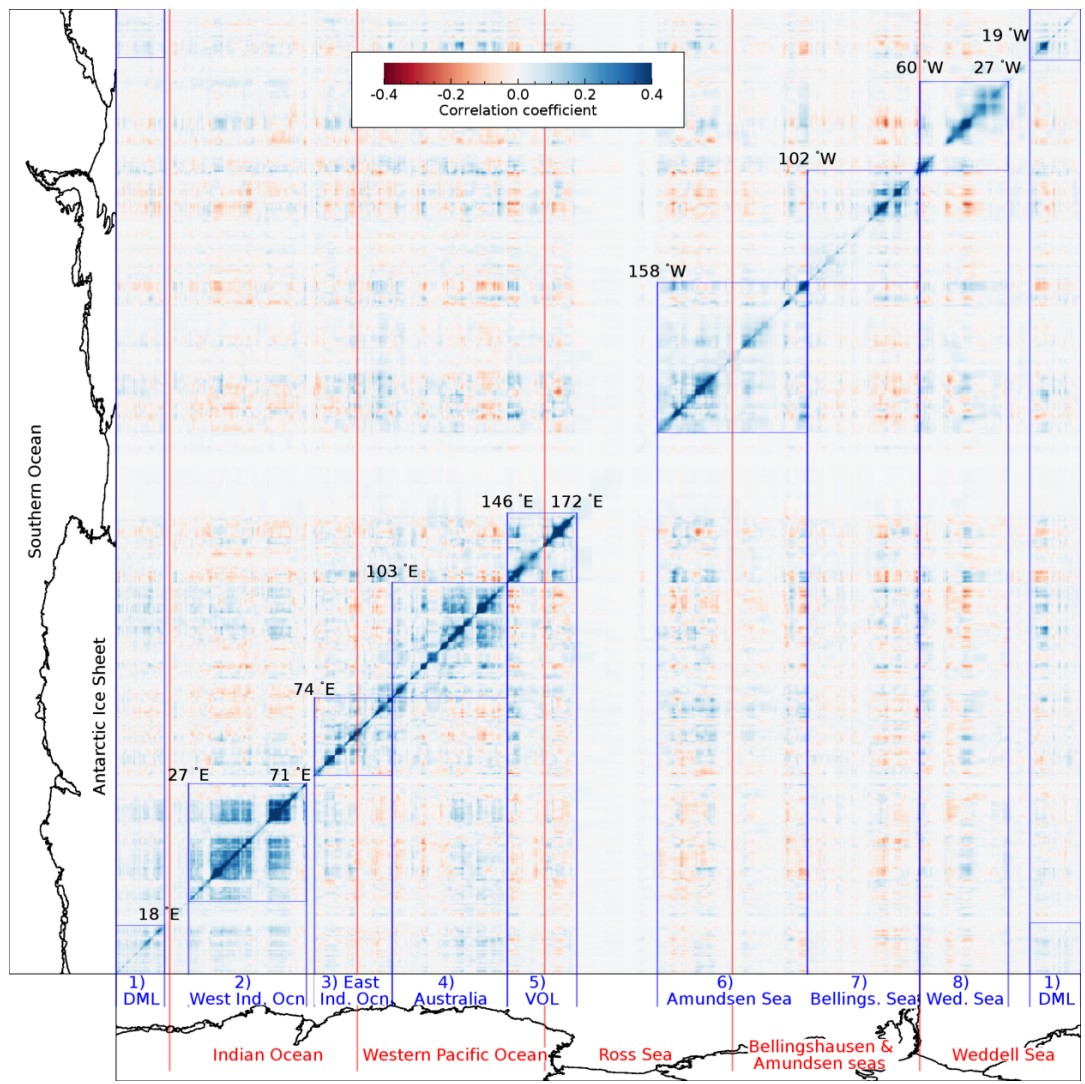

**Figure B1.** Fast ice anomaly cross-correlation matrix. Red vertical lines indicate the traditional sea ice sectors. Blue vertical lines and blue-outlined boxes highlight fast ice regions consisting of pockets of high cross-correlation, indicating regions which co-vary. DML is Dronning Maud Land and VOL is Victoria and Oates Lands.

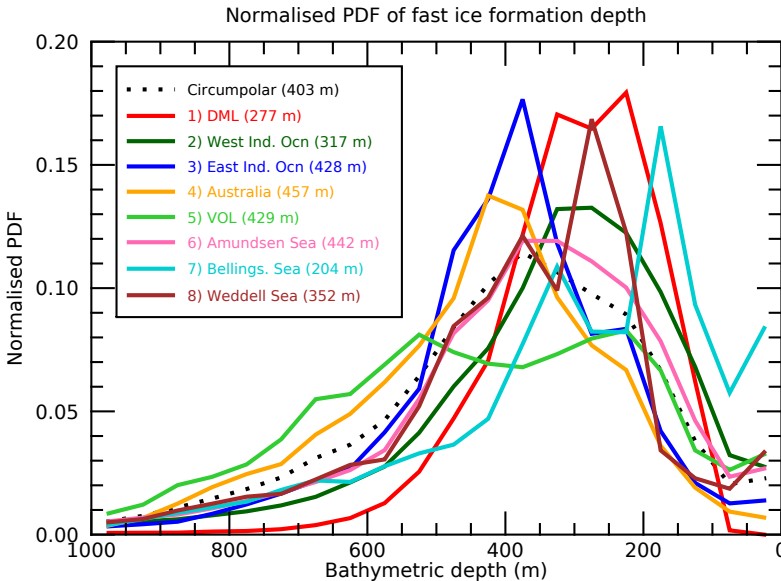

**Figure C1.** Normalised histograms of persistence-weighted fast ice formation depth for the eight fast ice regions. Mean formation depth for each region is indicated in the legend.

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
