# Peer review of "year record of circum-Antarctic landfast sea ice distribution allows detailed baseline characterisation, reveals trends and variability"

_The Cryosphere, 2021_

## Referee Comment (RC1)

This is a well-written and important manuscript that represents an in-depth baseline analysis of variability and change in circum-Antarctic fast-ice distribution. The analysis is based on interpreting a new 18-year (2000 – 2018) high-resolution fast ice extent time series generated from MODIS imagery that was released in 2020. It proposes eight new regions for the purpose of identifying trends in fast ice extent anomalies and fast ice persistence, as well as characterising the distribution of fast ice in relation to bathymetry. It also exploits a new technique to model the seasonal cycle of fast ice and sea ice extent that was presented in a manuscript by Handcock in Raphael in 2020.

I have only a few major comments / suggestions to make, followed by a number of minor comments and suggestions.

Major suggestions / comments (line numbers given where appropriate).

Line 75: The authors state that the new fast ice regions were grouped manually because an automated selection using a decorrelation length scale minimum-based approach (Raphael and Hobbs, 2014) did not work, and present their partitioning technique in Appendix B. However, sufficient details to determine how the partioning was done are not provided, other than stating that an investigation of the fast ice anomaly cross-correlation matrix as a function of longitude was undertaken. Can the authors provide more details on how this partioning was done, particularly as they are introducing a "fundamental new region definition". Can they also comment on the implications of using a non-conservative region definition (e.g. does not include all of the Antarctic coastline / ice shelf edges) as opposed to the Zwalley et al. (1983) oceanic sectors?

Lines 100–104: Can the authors provide a more indepth description of the technique they used to model the seasonal cycle of sea ice and fast ice. The Handcock and Raphael (2020) paper presents three techniques for modelling annual sea ice extent cycles that are time variant (amplitude only, phase only and amplitude + phase). These models were also only applied to daily sea ice extents, not 15-day interval fast ice extents, although it was noted in the Handcock and Raphael manuscript that this technique could readily be applied to other datasets.

Minor suggestions / comments (line numbers given where appropriate).

Lines 22 and 23: The first sentence of this paragraph is not a complete sentence, I suggest rewording by combining it with the next sentence.

Line 50: There is a missing "a" before "suitable underlying dataset".

Lines 76 and 77: "Raphael and Hobbs" is repeated.

Line 80: "the" is repeated.

Line 116: replace "approx" with "approximate".

Line 120: Be consistent with use of hyphenation with "mid" and "late".

Line 123: I do not understand what is meant by "(as a percentage of average residence time)". How is average residence time defined? If percentages are relative to an average time, why do they never exceed 100 %? This definition differs to the description in the Fig. 2 caption – the caption definition makes sense to me.

Line 143: Capitalise "coast".

Lines 160 + 161 (Comment only): Another area that experienced a large change from multi-year fast ice to seasonal fast ice in the period is the southern reaches of McMurdo Sound. This was due to the presence of large tabular icebergs (B-16 and C-16) (Brunt et al., 2006). I suspect the trend does not appear as stong here as in other regions due to the relative timing of the iceberg affected fast ice cover (2001 - 2011) with reference to the length of the data set (2000 - 2018).

Line 173: Replace reference to Fig. S3 with Fig. C1.

Line 176: Suggest moving "only" to before "useful".

Line 180: Replace "total sea ice extent" with "total fast ice extent".

Line 183: Remove duplicate "Fraser et al.".

Line 185: Replace Fig. 5b with Fig. 5c.

Line 186: Replace Fig. 5d with Fig. 5e.

Line 193: Replace "that" with "than".

Line 199: replace "than the that of sea ice" with "than that of sea ice"

Line 208: Replace "overall sea ice" with "the overall sea ice maximum".

Lines 223 and 226: Replace ref to Fig. S3 with Fig. C1.

Line 231: Replace Massom (2003); Massom et al. (2009) with (Massom, 2003; Massom et al., 2009)

Line 293: Suggest insert a comma after "however".

Lines 296 + 297: move $(0.67 \pm 0.55 \%/y)$ to before "sectors".

Line 321: What do the authors mean by "... was re-run using only pre-calving post-calving fast ice anomaly data."? I assume from the following sentences that the regional selection algorithm was run twice, once with pre-calving conditions, and a second time with post-calving conditions,

but this needs clarification.

Data availability. The authors need to add a description of how the sea ice concentration from the National Oceanic and Atmospheric Administration/National Snow and Ice Data Center Climate Data Record of Passive Microwave Sea Ice Concentration, Version 3 can be obtained, as well as a citation to Meier, W. N., F. Fetterer, M. Savoie, S. Mallory, R. Duerr, and J. Stroeve. 2017. NOAA/NSIDC Climate Data Record of Passive Microwave Sea Ice Concentration, Version 3. [Indicate subset used]. Boulder, Colorado USA. NSIDC: National Snow and Ice Data Center. doi: https://doi.org/10.7265/N59P2ZTG. [Date Accessed]. as described in the dataset's condition of use, reference: https://nsidc.org/data/G02202/versions/3.

Line 399: I could not find the Kooyman and Burns 1999 manuscript and Kooyman does not appear to list this publication on his website. I did find some other references to a 2009 publication in American Zoologist, so was left wondering if American Zoology should be American Zoologist?

Figure 1: I suggest the authors use the same y-axis label for sub-figure a and b. I find it confusing comparing the green line in sub-figure a with the green line in sub-figure b due to the different temporal scales between the two sub-figures, but I appreciate that too much detail might be lost if the width of sub-figure b was reduced.

Figure 2: "coast" in "Marie Byrd Land coast" needs to be capitalised. Missing 180° label.

Figure 5: Caption indicates that p-value of the trend is indicated in the title of each sub-plot, but I could not find this information in the sub-plot titles. To me the need for stating p-values in sub-titles is negated by the last sentence in the caption.

Figure A1: The trendline for the Indian Ocean sector is not easily distinguishable from the zero line. I suggest the authors consider using a colour other than black to represent the Indian Ocean anomalies and trend.

Figure B1: The vertical red and blue lines and blue boxes are somewhat difficult to view against the cross-correlation colour scale. I suggest either using thicker lines or choosing colours that do not fall within / near the cross-correlation colour scale. Acronyms for newly defined regions should be spelled out in the figure caption. I find it a bit confusing that the spatial scales on the two axes of a spatial cross-correlation plot are different, and that the coastline on the y-axis is facing the opposite direction relative to the plot than the coastline shown on the x-axis. It would also be useful to distinguish between land and ice shelves in the provided coastal outlines.

---

## Author Comment (AC1)

Reviewer 1: Dr Greg Leonard.

"This is a well-written and important manuscript that represents an in-depth baseline analysis of variability and change in circum-Antarctic fast-ice distribution."

- We thank Dr Leonard for his careful reading of the manuscript, and constructive comments which will improve it.
* * *
Major suggestions:

Line 75: The authors state that the new fast ice regions were grouped manually because an automated selection using a decorrelation length scale minimum-based approach (Raphael and Hobbs, 2014) did not work, and present their partitioning technique in Appendix B. However, sufficient details to determine how the partioning was done are not provided, other than stating that an investigation of the fast ice anomaly cross-correlation matrix as a function of longitude was undertaken. Can the authors provide more details on how this partioning was done, particularly as they are introducing a "fundamental new region definition". Can they also comment on the implications of using a non-conservative region definition (e.g. does not include all of the Antarctic coastline / ice shelf edges) as opposed to the Zwalley et al. (1983) oceanic sectors?

- This is a good point - this process could be made clearer. Reviewer 2's comment number three is also similar.
- We aim to select regions with high correlation within the region and low correlation outside of the region. As with the new region definition in Hobbs and Raphael (2014; hereafter H&R), this was largely a manual process, constrained by the content in Fig. B1. In the case of H&R, their selection was guided by functions of sea ice extent standard deviation and decorrelation length scale, i.e., two quantities whose boundaries did not always match spatially, necessitating a subjective decision. Furthermore, H&R select boundaries based on local minima of these quantities, however the choice of which local minimum should be selected, when more than one option exists (e.g., the boundary between the Ross and Bellingshausen/Amundsen regions in H&R Fig 1), is somewhat subjective. This parallels our selection and the subjective elements within. As with H&R, most section definitions here were quite objective (e.g., the Australia region: Fig B1 shows that this box contains only blue pixels, indicating positive cross-correlation within this region, and is surrounded by red pixels, indicating negative cross-correlation). However, we concede that for more gradually-decorrelating regions (e.g., the demarcation between the Eastern Indian Ocean and Western Indian Ocean regions), the subjective element is higher. We plan to address this in the manuscript by a) more clearly stating how these regions

were defined, and b) including discussion of some of this detailed comparison against H&R's similar region selection process.

- Regarding the reviewer's latter point on the non-conservative nature of this definition: It is worth elaborating on this point. Are there implications of this incomplete region definition for users? It's possible - especially if fast ice forms in the future where there currently is none. Currently, longitudes without regular/extensive fast ice are unable to be assigned a group, since it's not clear which of the two "neighbours" they should be assigned to. Under what scenario might fast ice exist there in the future? Probably only if a change in the distribution of large grounded icebergs was to interfere with the local ice-scape. It's not possible to determine which group those longitudes should be assigned without first seeing how their anomalies cross-correlate. In short, major ice-scape changes may both a) precipitate a need for reassigning these regions, or b) require longitudes currently "between" regions to be assigned a region. We plan to address this comment by emphasising these points in appendix B where region selection is discussed.

Lines 100–104: Can the authors provide a more indepth description of the technique they used to model the seasonal cycle of sea ice and fast ice. The Handcock and Raphael (2020) paper presents three techniques for modelling annual sea ice extent cycles that are time variant (amplitude only, phase only and amplitude + phase). These models were also only applied to daily sea ice extents, not 15-day interval fast ice extents, although it was noted in the Handcock and Raphael manuscript that this technique could readily be applied to other datasets.

- We agree this is not explained sufficiently, and apologise for that. We used the invariant annual cycle, that is, it is numerically the same year-to-year. This will be made clear around lines 100-104.
- The method can estimate the smooth cyclical spline based on an arbitrary and/or irregular data interval. We treated the fast ice extent value as if it was a point measurement on the day at the midpoint of the 15 day cycle. For example, if the start_doy was 61 and the end_doy was 75, we modeled it as if we had a single measurement at doy (61+75)/2. This will also be mentioned around lines 100-104.
* * *
Minor comments:

- Lines 22 and 23: The first sentence of this paragraph is not a complete sentence, I suggest rewording by combining it with the next sentence.
    - Good suggestion - will be combined.

- Line 50: There is a missing "a" before "suitable underlying dataset".
  - Thank you - will accept.
- Lines 76 and 77: "Raphael and Hobbs" is repeated.
  - Thank you - a hangover from a previous version.
- Line 80: "the" is repeated.
  - Will fix
- Line 116: replace "approx" with "approximate".
  - Will replace
- Line 120: Be consistent with use of hyphenation with "mid" and "late".
  - A search indicates that, depending on which result I read, "mid" should be hyphenated in places where "early" or "late" isn't, but I agree it looks a bit silly here. Will be fixed.
- Line 123: I do not understand what is meant by "(as a percentage of average residence time)". How is average residence time defined? If percentages are relative to an average time, why do they never exceed 100 %? This definition differs to the description in the Fig. 2 caption – the caption definition makes sense to me.
  - I agree it isn't worded very well. Reworded to match the caption definition.
- Line 143: Capitalise "coast".
  - Will fix
- Lines 160 + 161 (Comment only): Another area that experienced a large change from multi-year fast ice to seasonal fast ice in the period is the southern reaches of McMurdo Sound. This was due to the presence of large tabular icebergs (B-16 and C-16) (Brunt et al., 2006). I suspect the trend does not appear as stong here as in other regions due to the relative timing of the iceberg affected fast ice cover (2001 - 2011) with reference to the length of the data set (2000 - 2018).
  - Although your "comment only" may not need a detailed reply, I'm also intrigued as to why this region doesn't show a strong negative trend. If the iceberg-associated positive fast ice anomaly here occurred around the middle of the time series (i.e., around 2009) then I'd expect a minimal trend, but my understanding is this region experienced a positive anomaly much earlier than this (so the trend should be more strongly negative). I prepared a time series subset covering only the Sound (i.e., south of the Drygalski ice tongue):

[Figure]

The extensive fast ice cover throughout the 2016 winter appears to temper this expected negative trend.

- Line 173: Replace reference to Fig. S3 with Fig. C1.
  - Apologies for this oversight - will be fixed.
- Line 176: Suggest moving "only" to before "useful".
  - Good suggestion
- Line 180: Replace "total sea ice extent" with "total fast ice extent".
  - Yes - apologies!
- Line 183: Remove duplicate "Fraser et al.".
  - Will be fixed
- Line 185: Replace Fig. 5b with Fig. 5c.
  - Thank you!
- Line 186: Replace Fig. 5d with Fig. 5e.
  - Thanks again!
- Line 193: Replace "that" with "than".
  - Will be fixed
- Line 199: replace "than the that of sea ice" with "than that of sea ice"
  - Thank you - will be fixed.
- Line 208: Replace "overall sea ice" with "the overall sea ice maximum".
  - Will be fixed
- Lines 223 and 226: Replace ref to Fig. S3 with Fig. C1.
  - Thank you - will be fixed.
- Line 231: Replace Massom (2003); Massom et al. (2009) with (Massom, 2003; Massom et al., 2009)
  - Will be fixed.
- Line 293: Suggest insert a comma after "however".
  - I appreciate this attention to detail! Thanks for picking it up.
- Lines 296 + 297: move (0.67 ± 0.55 %/y) to before "sectors".
  - Good suggestion.
- Line 321: What do the authors mean by "... was re-run using only pre-calving post-calving fast ice anomaly data."? I assume from the following sentences that the

regional selection algorithm was run twice, once with pre-calving conditions, and a second time with post-calving conditions, 2 but this needs clarification.
- ○ Your interpretation is correct, and sorry this was ambiguous. There's an "and" missing here which wouldn't have helped. This will be rectified and clarified.
- Data availability. The authors need to add a description of how the sea ice concentration from the National Oceanic and Atmospheric Administration/National Snow and Ice Data Center Climate Data Record of Passive Microwave Sea Ice Concentration, Version 3 can be obtained, as well as a citation to Meier, W. N., F. Fetterer, M. Savoie, S. Mallory, R. Duerr, and J. Stroeve. 2017. NOAA/NSIDC Climate Data Record of Passive Microwave Sea Ice Concentration, Version 3. [Indicate subset used]. Boulder, Colorado USA. NSIDC: National Snow and Ice Data Center. doi: https://doi.org/10.7265/N59P2ZTG. [Date Accessed]. as described in the dataset's condition of use, reference: https://nsidc.org/data/G02202/versions/3.
  - ○ Apologies for this oversight - Reference and description will be added.
- Line 399: I could not find the Kooyman and Burns 1999 manuscript and Kooyman does not appear to list this publication on his website. I did find some other references to a 2009 publication in American Zoologist, so was left wondering if American Zoology should be American Zoologist?
  - ○ Yes - very well spotted! Will change to "American Zoologist".
- Figure 1: I suggest the authors use the same y-axis label for sub-figure a and b. I find it confusing comparing the green line in sub-figure a with the green line in sub-figure b due to the different temporal scales between the two sub-figures, but I appreciate that too much detail might be lost if the width of sub-figure b was reduced.
  - ○ Yes the y-axis labels should be the same - will rectify. We'll try to resist compressing the width of Fig 1b for the reason you stated, but will explain this in the caption.
- Figure 2: "coast" in "Marie Byrd Land coast" needs to be capitalised.
  - ○ Will rectify
- Missing 180° label.
  - ○ Will add
- Figure 5: Caption indicates that p-value of the trend is indicated in the title of each sub-plot, but I could not find this information in the sub-plot titles. To me the need for stating p-values in sub-titles is negated by the last sentence in the caption.
  - ○ Apologies - we replaced the reporting of p-values (in an earlier iteration of the figures) with confidence intervals. As you stated, since all are significant, p-values are redundant. Will amend the caption to refer to confidence intervals only, rather than confidence intervals and p-values.
- Figure A1: The trendline for the Indian Ocean sector is not easily distinguishable from the zero line. I suggest the authors consider using a colour other than black to represent the Indian Ocean anomalies and trend.
  - ○ Good suggestion - we will change the colour.

- Figure B1: The vertical red and blue lines and blue boxes are somewhat difficult to view against the cross-correlation colour scale. I suggest either using thicker lines or choosing colours that do not fall within / near the cross-correlation colour scale. Acronyms for newly defined regions should be spelled out in the figure caption. I find it a bit confusing that the spatial scales on the two axes of a spatial cross-correlation plot are different, and that the coastline on the yaxis is facing the opposite direction relative to the plot than the coastline shown on the x-axis. It would also be useful to distinguish between land and ice shelves in the provided coastal outlines.
    - Thanks for the suggestions. We will use thicker lines for the region boundaries and define the acronyms used in the caption. The spatial scales on the two axes are the same, and the coastline of the y-axis is facing the correct way (at least, it's the intuitive way for me: if one rotates the page 90 degrees clockwise, e.g., to read the y-axis label, then the coastline orientation (on the y-axis) is the same as that on the x-axis without rotation), but happy to discuss further if I've missed something. We will add "Southern Ocean" and "Antarctic Continent" to the y-axis spatial figure to remove any ambiguity (I don't think these will fit onto the x-axis spatial figure).

---

## Author Comment (AC2)

Reviewer 2

"This submission explores many aspects of fast ice around the periphery of Antarctica, including trends, links with bathymetric depth, monthly timings of minimum and maximum coverage, age, and persistence etc. It is based on the analysis of 18 years of record obtained via hires remote sensing.
The submission has the potential to make a significant contribution to the literature, but it is not quite there yet. Before I would be able to recommend acceptance, there are a number of issues which need to be addressed."

- We thank Reviewer 2 for their careful reading of the manuscript, and constructive suggestions of ways to improve it.
* * *
● Lines 51-54: The authors should comment on how their dataset might different from one that used the improved scheme of Paul and Huntemann applied to MODIS to detect cloud cover over Antarctic sea ice as well as its discrimination from sea-ice cover and open-water areas. See Stephan Paul and Marcus Huntemann, 2021: Improved machine-learning-based open-water-sea-ice-cloud discrimination over wintertime Antarctic sea ice using MODIS thermal-infrared imagery. The Cryosphere, 15, 1551-1565, doi: 10.5194/tc-15-1551-2021.
  ○ Thank you for mentioning this manuscript, which is already of great interest to us as a way of providing an independent (to the widely-used MOD35 MODIS product) cloud mask. While the Paul and Huntemann dataset is focused on retrieval of polynya extent, the dataset used in this paper focuses on fast ice extent. These two often share a common boundary. We are already exploring overlap between our data processing and hope to progress this in the future. We will provide comment on this around lines 51-54.
● Lines 56-58, and various other places in paper: It makes sense that the five sectors that have been traditionally used for sea ice analyses might not be appropriate for fast ice (because of the very different dynamics and thermodynamics). The authors later go on to explain how they choose eight coherent sectors for the fast ice investigation. Then throughout the paper they refer to the 'newly-defined regions'. This becomes a little tedious and is really unnecessary – I suggest they just refer to the 'regions', as the meaning will be quite clear.
  ○ A good suggestion - thank you. We will incorporate this.
● In the Appendices they describe the autocorrelation approach to identifying the eight regions, and their approach is reasonable. They also point out some of the caveats of how they have done this. An extra caveat that should be mentioned is that they have obtained these semi-coherent sectors in a 'a posteriori' fashion rather from, e.g., physical arguments. Hence, by its very nature coherent collections are identified, which

in turn will mean than any trends will have an enhanced level of statistical significance. I don't have a great problem with the approach, but the caveat should be made clear.
- ○ The reviewer is correct: these are certainly defined a posteriori (as with Raphael and Hobbs (2014)). A caveat along these lines will be added in Appendix B.
- Appendix A and B: As a point related to that raised immediately above, it seems that the 'traditional' five sectors are only used in these Appendices (and in Fig. A1). Given that, I strongly suggest that these five sectors (the red ones) be removed from Figure 2, as it makes that Fig. more complicated than it needs to be. (The longitude limits of these sectors can be presented in the text in the Appendices). As a separate idea I would suggest that an extra column be added to Table 1 to present the trends (and p values) for the eight regions; this would mean that the results of the trend analyses (which are perhaps the most interesting aspect of the paper) are presented 'front and center' to catch the reader's eye.
  - ○ We agree, and are very much in favour of de-cluttering Fig. 2 in any way possible. This (removing the red lines/sectors from Fig 2) is a good suggestion. We propose moving the sector boundaries to the caption of Fig. A1. The column addition to Table 1 is a great suggestion - however, since the p-values for the regions are all similar (and very close to zero), we will report the confidence interval instead of the p-value.
- Also on Figure 2 (and also Figures 3 and 4) the sectors are labelled 1 thru 8. These numbers are not referred to, so they should be deleted.
  - ○ Will be deleted on these three figures - well picked up, thank you.
- Line 76-77: Author names have been repeated.
  - ○ Will be rectified.
- Lines 83-86: Why not simply say here 'the first four Fourier components'? Also, perhaps make clear why the standard method of calculating Fourier amplitudes and phases was not used. What was the advantage of the L-M approach here?
  - ○ Will change to "first four components". The L-M implementation (mpfit in IDL) was used here due to prior experience with this package, and provides no advantage to the traditional approach other than speed of execution (important on this grid of ~11 billion points (5625*4700 spatial; 432 temporal)). This will be mentioned in lines 83-86.
- Line 95 (and elsewhere where relevant): Jodie Smith's paper now has a doi, so should now be referenced as 'Smith et al. (2021)'
  - ○ Thank you for providing this update.
- Line 113: To avoid any possible confusion perhaps best to be explicit and write 'circumpolar extent time' as 'circumpolar fast ice extent time'.
  - ○ Will be modified.
- Line 117: Also cite here the update of Simmonds et al., 2021: Trends and variability in polar sea ice, global atmospheric circulations and baroclinicity. Annals of NY Acad. Sciences, doi: 10.1111/nyas.14673.
  - ○ Interesting reference - will be added, thank you.

- Lines 128-132: Authors should comment on the physics that may be responsible for these phase difference between the fast ice and the sea ice.
  - This comment is already given in the discussion. We feel that it fits better here (lines 198-207).
- Line160: Change 'Fogwill et al. (2016),' to '(Fogwill et al., 2016),'
  - Will be rectified
- Lines before lines 162 (caption of Figure 5): The caption text says '… trend p-value are indicated in the title of each sub-plot' but the p values are not shown (only the standard error). Please add the p values.
  - Apologies - this mistake was also picked up by Reviewer 1 (Dr Greg Leonard). We will rectify this mistake as he suggested.
- Lines 162-164: Authors should present some ideas or speculation on what this environmental forcing over a large spatial scale' might be.
  - Very happy to speculate on this - but it's purely speculation (at this stage). Given the widespread distribution of the positive trend along the eastern part of the Weddell Gyre, we speculate that this environmental association is likely oceanic in nature. Following careful reading of Simmonds and Li (2021), we also note that this is a region of increasing summertime, springtime and wintertime sea ice concentration (their Fig. 2), which may favour formation of more extensive or longer duration of fast ice coverage, i.e., this fast ice trend may be associated with an oceanic trend which has atmospheric drivers. This will be added around lines 162-164.
- Line 183: Delete one of the 'Fraser et al.'.
  - Will rectify.
- Lines 180-191: Some interesting points are raised here, and certainly warrant future work. For the moment however, some extra discussion is required here. The tentative link/association made here with sea ice extent should be reinforced with comparison with sea ice concentration. The four regions of fast ice decrease and four of increase for the most part closely follow the trends in sea ice concentration shown by Li et al. (2021) 'Trends and variability in polar sea ice, global atmospheric circulations and baroclinicity. Ann. NY Acad. Sci., doi: 10.1111/nyas.14673'. Reference to that paper and some extra comments on this will make this part of the interpretation much stronger.
  - Thank you again for bringing this paper to our attention. It's an important point (to link the observed fast ice trend distribution to that of sea ice concentration, in addition to the extent discussion presented here). While the time periods of the Simmonds and Li paper differ to that presented here, we can indeed see that the general distribution of trend coincides with the southern hemisphere trend in concentration from 1979 to 2020 in Sept-Nov (Fig. 2B, bottom row), but this link is somewhat less convincing for other seasons. That this relationship is strongest is Sept-Nov is not surprising, since this season coincides with maximum fast ice extent. We will add this discussion around lines 180-191.

- Lines 199-207: Also helpful to reference at this point in the text the more recent paper by Aoki, S., 2017: Breakup of land-fast sea ice in Lützow-Holm Bay, East Antarctica, and its teleconnection to tropical Pacific sea surface temperatures. Geophysical Research Letters, 44, 3219-3227, doi: 10.1002/2017GL072835.
  - A good suggestion. Although the precise mechanism was not discussed in the Aoki paper, if the mechanism involved an oceanic link, then this might also explain why fast ice lags sea ice. This discussion will be added around line 207.
- Lines 283-285: Relevant remote atmospheric teleconnections into the Peninsula from warming down in the Tasman Sea have recently been identified by Sato, K and coauthors 2021 - Antarctic Peninsula warm winters influenced by Tasman Sea temperatures. Nature Comms, 12, 1497, doi: 10.1038/s41467-021-21773-5. Valuable to also reference this here.
  - A good point: the Sato et al paper is a good example of a remote teleconnection which has demonstrable impacts on relevant (for fast ice) local climate parameters, even though it does not specifically mention fast ice. Will be incorporated around line 285.
- Line 317: Reinforce this statement by also referencing Sato et al., 2021 ('Antarctic skin temperature warming related to enhanced downward longwave radiation associated with increased atmospheric advection of moisture and temperature. Env. Res. Lett., 16, 064059, doi: 10.1088/1748-9326/ac0211') (his Fig. 4).
  - Reference (and Fig. 4 in particular) is highly relevant and will be added.
- Lines 395-396: Please to note missing details plus wrong order of authors - Kim, M., H.-C. Kim, J. Im, S. Lee and H. Han, 2020: Object-based landfast sea ice detection over West Antarctica using time series ALOS PALSAR data. Remote Sensing of Environment, 242, 111782, doi: 10.1016/j.rse.2020.111782.
  - Apologies - will be rectified
- Lines 450-451: Please note full details of this paper are Claire L. Parkinson, 2019: A 40-y record reveals gradual Antarctic sea ice increases followed by decreases at rates far exceeding the rates seen in the Arctic. Proceedings of the National Academy of Sciences of the United States of America, 116, 14414-14423, doi: 10.1073/pnas.1906556116.
  - Will be rectified.
- Lines 461-462: Updated details are: Smith J., Nogi Y., Spinoccia M., Dorschel B. and Leventer A. (2021) A bathymetric compilation of the Cape Darnley region, East Antarctica. Antarc. Sci., doi: 10.1017/S0954102021000298.
  - Thank you for the update - will be updated.
- Lines 466-467: Turner, John et al., 2016: Absence of 21st century warming on Antarctic Peninsula consistent with natural variability. Nature, 535, 411-415, doi: 10.1038/nature18645.
  - Will be completed.

---

## Referee Report (RR1)

**A review of The Cryosphere manuscript**
**tc-2021-121-manuscript-version3 by Fraser et al.**

I consider the manuscript improved overall and thank the authors for their sincere efforts in addressing the review comments from the first round.

I have only the few minor comments below to make.

Line 10: Change "eastern" to "western".

Figure B1: My apologies for my earlier comment about different spatial scales for the x and y axis – this was a "screen" effect. When I printed out the figure (which I should have done originally), I confirmed that the spatial scales are the same.

Lines 500 – 501: The Paul and Huntemann reference that was added after the first review is to The Cryosphere Discussions manuscript (doi: 10.5194/tc-2020-159), instead of The Cryosphere manuscript (doi: 10.5194/tc-15-1551-2021).

---

## Author Response (AR2)

Dear Dr Howell,

Thank you for your quick work in pregressing our manuscript.

We thank both reviewers for their careful check of our changes, and pointing out a few more deficiencies. These have now been rectified as suggested by the examiners (details below in bold).

Kind regards on behalf of the author team.
Alex.

Reviewer 1:
Line 214: Change 'minimum, (Eayrs' to 'minimum (Eayrs'
**Changed - thank you.**

Line 262: Change 'as detailed in (Li et al., 2020).' to 'as detailed in Li et al. (2020).'
**Changed - thank you.**

Line 395: Some of the bibliographical info is repeated on this line.
**This has been corrected.**

Lines 426-427: ALL the reference details of this paper should be presented. A similar comment applies to quite a few of the other references in the paper – please check carefully.
**We have gone through the reference list and added any parts we found missing.**

Reviewer 2:
Line 10: Change "eastern" to "western".
**Thank you for picking up on this embarassing mistake! Amended**

Figure B1: My apologies for my earlier comment about different spatial scales for the x and y axis – this was a "screen" effect. When I printed out the figure (which I should have done originally), I confirmed that the spatial scales are the same.
**Understood - thank you.**

Lines 500 – 501: The Paul and Huntemann reference that was added after the first review is to The Cryosphere Discussions manuscript (doi: 10.5194/tc-2020-159), instead of The Cryosphere manuscript (doi: 10.5194/tc-15-1551-2021).
**Updated correctly now - thank you.**